# Diagnostic and Psychometric Properties of the Arabic Sensory Processing Measure—Second Edition, Adult Version

**DOI:** 10.3390/jcm14103283

**Published:** 2025-05-08

**Authors:** Hind M. Alotaibi, Ahmed Alduais, Fawaz Qasem, Muhammad Alasmari

**Affiliations:** 1Department of English Language, College of Language Sciences, King Saud University, Riyadh 11586, Saudi Arabia; 2Department of Psychology, Norwegian University of Science and Technology, NO-7491 Trondheim, Norway; 3Department of English Language & Literature, College of Arts of Letters, University of Bisha, Bisha 67714, Saudi Arabia; faqasem@ub.edu.sa (F.Q.); moaalasmri@ub.edu.sa (M.A.); 4King Salman Center for Disability Research, Riyadh 11614, Saudi Arabia

**Keywords:** Arabic validation, confirmatory factor analysis, early detection, internal consistency, Saudi Arabia, sensory processing (disorders), SPM-2

## Abstract

**Background:** Sensory processing difficulties can interfere with daily functioning and participation across adulthood. While standardized assessment tools exist, culturally validated instruments for Arabic-speaking adults remain limited. **Objectives**: This study aimed to validate the Arabic version of the Sensory Processing Measure—Second Edition (SPM-2) Adult Self-Report form in a Saudi population and evaluate its utility for the early detection of sensory processing challenges in at-risk individuals. **Methods**: A total of 399 Saudi adults (205 females and 194 males), aged 21 to 87 years (M = 44.1; SD = 16.2), completed the Arabic SPM-2 online. The scale consists of eight subscales, six of which form the Sensory Total score—Vision, Hearing, Touch, Taste and Smell, Body Awareness, and Balance and Motion—representing core sensory processing abilities (i.e., Sensory Total (ST)). The remaining two—Planning and Ideas and Social Participation—capture higher-order integrative functions and do not contribute to the ST. **Results**: The overall scale demonstrated strong internal consistency (α = 0.89), with subscale alphas ranging from 0.43 (Hearing) to 0.70 (Body Awareness). Confirmatory factor analysis (CFA) (χ^2^ [3052] = 4147.4; *p* < 0.001) showed good absolute fit (RMSEA = 0.030) and moderate incremental fit (CFI = 0.74; TLI = 0.73), values that are typical for large-df models. Descriptive and cluster analyses identified distinct participant subgroups with elevated frequency ratings (scores of 2 or 3) suggestive of sensory risk. Significant age-related differences were observed across multiple sensory domains, while no significant sex-related effects were found. **Conclusions**: Although Social Participation and Hearing showed lower reliability, the Arabic SPM-2 exhibits sound internal structure and therefore shows promise for future clinical application once criterion validity is established. The findings support its application in culturally responsive screening, early risk identification, and intervention planning in Arabic-speaking contexts.

## 1. Introduction

Sensory processing disorder (SPD) refers to the inefficient registration, modulation, or organization of sensory input that produces maladaptive behavioral and functional responses [1]. Epidemiological reviews place SPD in 5–16% of typically developing children, while studies of autism spectrum disorder (ASD) report sensory difficulties in up to 90% of cases [2,3,4]. Clinical manifestations range from tactile defensiveness and auditory filtering problems to proprioceptive inaccuracy, each of which can undermine play, learning, and social participation [5,6,7].

Untreated sensory challenges frequently persist into adolescence and adulthood, predicting poorer quality of life, heightened anxiety, reduced occupational performance, and restricted social participation [8,9]. Early, targeted sensory-based interventions that combine environmental modification, graded sensory experiences, and caregiver coaching have been shown to improve adaptive behavior, self-regulation, and daily functioning [10,11,12,13].

Despite growing awareness, timely identification remains difficult. Caregivers may mis-label hypersensitivity, hyposensitivity, or sensory-seeking behaviors as conduct problems, and professionals often lack brief, culturally responsive screening instruments [14,15,16]. As a result, many children are referred only once academic demands accentuate sensory difficulties, prolonging treatment delay [17]. Community surveys further show that in Arabic-speaking settings, large proportions of affected individuals remain unrecognized, emphasizing the value of psychometrically sound tools adapted for local culture [18,19].

Conceptual models guide both assessment and intervention. Dunn’s Sensory Processing Framework maps neurological threshold against behavioral self-regulation, producing four patterns—seeking, avoiding, sensitivity, and registration—that characterize individual reactivity [20,21]. Dunn’s framework conceptualizes sensory processing along two continua—neurological threshold and behavioral self-regulation—that intersect to produce four patterns: seeking, avoiding, sensitivity, and registration [20,21]. In this model, the neurological-threshold continuum reflects how much sensory input is required for a neuron to fire: people with low thresholds notice stimuli readily, whereas those with high thresholds need stronger input. The behavioral self-regulation continuum ranges from passive (allowing sensations to occur) to active (strategically controlling exposure) [20,21]. Miller’s taxonomy separates sensory modulation, sensory-based motor, and sensory discrimination disorders, broad headings that encompass specific functional deficits [22,23]. An ecological perspective further emphasizes the interaction between personal sensory style and contextual demands [24]. These constructs underpin standardized measures such as the original Sensory Processing Measure (SPM) and its classroom variants [25], and inform evidence-based treatment protocols [26,27].

Assessment typically combines multiple methods. Report questionnaires capture behavior across home, school, and community [28,29,30,31], whereas the Sensory Integration and Praxis Tests permit direct observation [32,33,34]. Differential diagnosis is challenging because sensory signs overlap autism, attention-deficit hyperactivity disorder, and other neurodevelopmental problems [35,36]. Moreover, sensory profiles vary with age, culture, and temperament [37,38,39], necessitating norm-referenced, culturally sensitive instruments.

Measurement options have proliferated. The Sensory Profile suite—spanning infancy to older adulthood—remains a mainstay [14,40]. Newer tools such as the Sensory Processing and Self-Regulation Checklist and the Evaluation in Ayres Sensory Integration integrate advances in neuroscience and modern psychometrics [41,42]. Meta-analyses report robust reliability for most measures [43,44], but cross-cultural validation often lags behind translation efforts [45,46]. Reviews of Arabic versions note gaps in normative data and factorial evidence [18,19], reinforcing calls for rigorous adaptation studies.

A recent advance is the Sensory Processing Measure—Second Edition (SPM-2), which extends the original SPM across five age levels and multiple environments [25,47]. The Adult Self-Report form yields eight domain scores—Vision, Hearing, Touch, Taste–Smell, Body Awareness, Balance–Motion, Praxis, and Social Participation—and an overall Sensory Total. Initial English studies demonstrate high internal consistency (α = 0.79–0.95), excellent test–retest reliability (r = 0.89–0.97), and sensitivity for discriminating ASD, ADHD, and learning disabilities [47]. Bibliometric mapping shows the tool’s rapid uptake in occupational therapy, psychology, and education [48], and content analyses confirm that its items align closely with sensory modulation constructs [27].

Translation alone cannot ensure conceptual equivalence; forward–backward translation, expert reconciliation, and empirical testing are essential to preserve construct validity [49,50]. Culture shapes both sensory experiences and caregiver reporting [51], as illustrated by adaptations of the Sensory Profile in Taiwan, Israel, and Turkey [38,39,52]. Within the Arab world, the psychometric evidence base is thin: Arabic versions of the Adolescent/Adult Sensory Profile show acceptable reliability but unresolved factor structure [53], and Preschool/Home SPM forms exhibit subscale weaknesses [54]. Saudi epidemiological data further indicate gender and regional variations in sensory disorder prevalence, with higher severe Hearing issues in females and elevated Balance–Motion deficits in rural areas [55]; stigma and variable parental expectations may also influence reporting [56]. Several Arabic translations of sensory tools exist, yet each has important constraints.

The Arabic Sensory Profile-2 versions were validated only with children and relied on convenience samples < 200 without factor-analytic evidence [18]. Preschool and Home forms of the Arabic SPM demonstrated adequate reliability, but the validation stopped at age 12 and reported two subscales with α < 0.60 [57]. The Adolescent/Adult Sensory Profile translation showed acceptable internal consistency (α = 0.70–0.76) but an unstable factor solution and no normative cut-offs for adults [53]. None of these tools evaluated measurement invariance across sex or age, and no Arabic instrument currently offers population norms for adults. These gaps limit clinical decision-making and underscore the need for a psychometrically robust, adult-specific Arabic measure—precisely the aim of the present study.

Accordingly, the present study translated and culturally adapted the SPM-2 Adult Self-Report form into Arabic and evaluated its internal consistency, construct structure and diagnostic accuracy in a community sample of Saudi adults aged 21–87 years. We also examined sex- and age-related score patterns and generated preliminary norms to facilitate early, culturally sensitive identification and intervention. Following International Test Commission guidelines, we used forward–backward translation, expert review, cognitive debriefing, and large-scale field testing to maximize linguistic clarity and cultural relevance. Internal consistency was tested with Cronbach’s alpha, construct validity with CFA of the expected eight-factor model, and diagnostic accuracy by comparing frequency ratings that indicate sensory risk clusters. To our knowledge, this is the first study to validate the Adult Self-Report form of the SPM-2 in Arabic, extending earlier Arabic adaptations that were limited to Preschool or Child versions [57]. An adult, culture-specific sensory assessment supports the differential diagnosis of neurodevelopmental and functional neurological conditions and guides occupational-therapy intervention in primary and specialist care [58]. Therefore, this study aimed (i) to translate and culturally adapt the Adult SPM-2 into Arabic, and (ii) to evaluate its internal consistency, structural validity, and diagnostic accuracy in a community sample of Saudi adults.

## 2. Methods

### 2.1. Sample

The study utilized volunteer sampling, conducted by disseminating the online Arabic version of the SPM-2 across various colleagues in higher education institutions and enterprises beyond the higher education sector. Volunteer sampling offers practical advantages, such as ease of implementation, cost-effectiveness, and rapid data collection [59]. However, it also has notable weaknesses, including potential sampling bias and limited generalizability due to the self-selection of participants [60]. To minimize these limitations, the distribution targeted diverse settings to achieve a broader representation of participants. Table 1 illustrates the demographic characteristics of the study’s participants, categorized by sex and age group.

An a priori power analysis (G*Power 3.1, χ^2^ test; w = 0.20; α = 0.05; 1 − β = 0.95) indicated a minimum of 305 participants; our final N = 399 exceeded this threshold. Sampling: non-probability volunteer sampling stratified by decade (21–30 years, … 81–87 years) was used to approximate the Saudi adult age distribution. Inclusion criteria: (i) native Arabic speaker; (ii) 21–87 years; (iii) residence in Saudi Arabia; (iv) completion of all 80 items; (v) no self-reported psychiatric or neurological diagnosis. Exclusion criteria: age < 21 years or failure to meet any inclusion criterion.

Overall means of responses across the 80 items ranged from 1.29 (SD = 0.70) among young adult females to 1.51 (SD = 0.74) among older adult males. These scores indicate mild-to-moderate levels of perceived sensory difficulty. Older participants, particularly males, reported higher sensory processing difficulty scores compared to younger groups, suggesting an increase in perceived sensory issues with age. Standard deviations were relatively consistent across all groups, demonstrating stable variability in sensory processing responses regardless of sex or age category. All individuals in the study sample were from Saudi Arabia, ensuring cultural homogeneity, and none reported a clear history of psychiatric conditions. This selection criterion was applied to minimize potential confounding effects of psychiatric comorbidities on sensory processing results, enhancing the validity and generalizability of the findings within the Saudi adult population.

### 2.2. Design

This research was a cross-sectional psychometric validation in which the Arabic SPM-2 and demographic data were collected once from each participant. In epidemiologic terminology, a study is cross-sectional when measurement occurs at a single point in time, irrespective of whether probability sampling or prevalence estimation is undertaken [61,62]. However, this design limits the ability to infer causality or to observe changes over time [61]. This study adhered to the International Test Commission (ITC) Guidelines on Test Use (https://www.intestcom.org/), which include principles such as ensuring fairness, minimizing bias, ensuring accessibility, and maintaining accuracy in test scoring and interpretation. Because the objective was to evaluate measurement properties rather than population prevalence, we followed the COSMIN reporting guideline for studies on measurement instruments (https://www.cosmin.nl, accessed on 26 April 2025) [63].

The translation of psychometric scales involves several critical issues as outlined in the ITC guidelines. Ensuring linguistic and cultural equivalence is vital to maintain the construct validity across populations. Key challenges include differences in language structure, cultural context, and item interpretation, which may alter the meaning or difficulty of test items. Selecting qualified translators who understand the target culture and construct being measured is crucial, as well as employing robust translation designs like forward–backward methods or reconciliation procedures. Additionally, the adapted version must be empirically validated to confirm reliability and equivalence while addressing format, scoring, and administration differences. The documentation of the process is often overlooked but remains essential for transparency and credibility. Overall, the goal is to achieve a test adaptation that is functional, culturally appropriate, and psychometrically sound.

To achieve these guidelines, this study employed careful translation and cultural adaptation procedures, rigorous expert reviews, detailed item analyses, and consistent psychometric evaluations (e.g., reliability and validity testing). Ethical practices were strictly followed, including informed consent, confidentiality, and voluntary participation, thereby strengthening the methodological rigor and ethical integrity of the research.

### 2.3. Measures

**Sensory Processing Measure—Second Edition:** The primary instrument employed was the SPM-2, specifically adapted to Arabic. The original SPM-2 is a widely validated psychometric tool used to assess sensory processing across multiple sensory domains. The SPM-2 includes forms for five age levels, namely, Infant/Toddler, Preschool, Child, Adolescent, and Adult, which can be independently or jointly used to evaluate sensory functioning comprehensively. Each form consists of 80 Likert-type items rated based on the frequency of specific behaviors (Never, Occasionally, Frequently, Always). This study specifically utilized the Adult Age Level version (ages 21–87), focusing on the Self-Report form completed by the adults themselves [47].

The original (English) SPM-2 shows excellent internal consistency. For the Self-Report form, Sensory Total reliability is α = 0.94–0.96, with individual domains such as Hearing, Body Awareness, and Balance/Motion in the range α = 0.81–0.90; Planning and Ideas remains strong (α = 0.84–0.86), whereas Touch and Social Participation are moderate (α = 0.73–0.80). For the Rater form, Sensory Total is α = 0.92–0.98, and most domains fall in the range α = 0.83–0.91; Taste/Smell performs well (α = 0.84–0.89) while Touch is more variable (α = 0.62–0.88). These values, reported across age groups, confirm the robust reliability of the SPM-2—particularly for the composite score—and provide a benchmark for the present Arabic validation [47].

### 2.4. Procedure

**Data Collection:** The study protocol received ethical approval from the Institutional Review Board (IRB) of King Saud University, Riyadh, Saudi Arabia (Ref. No.: KSU-HE-24-1093) on 12 December 2024. Following IRB approval, the Arabic version of SPM-2 was administered online. Participants provided informed consent electronically, voluntarily agreeing to partake in the research. The survey requested minimal demographic information (sex, age, and living area) to ensure anonymity and maintain confidentiality. Participants confirmed that they had no known psychiatric conditions, reducing potential confounding factors.

**Data Analysis:** Descriptive statistics summarized demographic characteristics and overall sensory processing difficulty scores. Reliability was assessed using Cronbach’s alpha to determine internal consistency across the scale and individual subscales. Item–total correlations were calculated to evaluate item coherence within subscales. Validity analysis encompassed content, construct, and structural validity, employing inter-scale correlations and CFA to confirm the factor structure proposed by Ayres Sensory Integration theory. A two-way ANOVA assessed the impact of sex and age category on sensory processing scores, supplemented by cluster analysis to identify subgroups potentially at risk of SPDs. All statistical analyses were performed using *Jamovi* (Version 2.3.26), Sydney, Australia and *Python* 3.13.1, the Netherlands, with significance levels set at *p* < 0.05 [64,65]. Statistical tools included various *R packages*, *Australia*, integrated within *Jamovi*, such as psych for reliability analyses [66], *lavaan* for confirmatory factor analysis [67], *semPlot* for visualizing CFA results [68] car for assumption checks and linear modeling support [69], factoextra for visualizing multivariate analyses [70], and jamovi modules such as *GAMLj* for general linear modeling [71] and *snowCluster* for cluster analysis [72]. Between-group analyses were restricted a priori to age category and sex—the two demographic factors reported in the original SPM-2 norms—so that statistical power was preserved and results could be directly compared with the source instrument.

## 3. Results

Consistently with our twofold aim, (i) to verify the psychometric soundness of the Arabic SPM-2 and (ii) to illustrate its clinical utility, the findings are presented in five successive steps: reliability, validity, confirmatory factor structure, demographic influences, and diagnostic capability. This order moves from basic internal consistency through higher-order construct verification to evidence of how the scale flags adults at potential sensory risk.

### 3.1. Reliability Analysis

Reliability was assessed using Cronbach’s alpha, a standard measure for evaluating internal consistency (See Table 2). The reliability analysis for the overall SPM-2 scale demonstrated high internal consistency, with a Cronbach’s alpha of 0.89 (M = 1.3784; SD = 0.24789). Item–total statistics indicated good overall consistency, as most items showed moderate item–rest correlations ranging from 0.18 to 0.49. Specifically, Item 39 exhibited the highest item–rest correlation (0.49), contributing significantly to the overall reliability. A few items displayed notably lower correlations, such as Item 12 (0.02) and Item 17 (0.04), suggesting limited contribution to the overall scale reliability. Despite these isolated lower correlations, the overall reliability remained robust, indicating that the SPM-2 is a reliable measure for assessing sensory processing among adults. Among the subscales, Body Awareness demonstrated the highest reliability (α = 0.70), while Social Participation had notably low reliability (α = 0.18). Vision (α = 0.55), Hearing (α = 0.43), Touch (α = 0.58), Taste and Smell (α = 0.52), Balance and Motion (α = 0.58), and Planning and Ideas (α = 0.61) displayed moderate internal consistency.

### 3.2. Validity Analysis

Validity refers to how accurately a measure assesses the concept it is intended to measure, ensuring that the results genuinely reflect the underlying construct. To thoroughly assess the validity of the Arabic SPM-2, three main types of validity were evaluated: content validity, construct validity, and structural validity. Content validity was established by carefully designing the Arabic SPM-2 items to reflect all aspects of sensory processing, following rigorous expert review, careful item selection, and refinement procedures consistent with the original SPM-2 guidelines. Construct validity, demonstrating that the SPM-2 accurately captures sensory processing constructs as theorized by Ayres Sensory Integration theory—including visual, auditory, tactile, olfactory/gustatory, proprioceptive, and vestibular systems, as well as praxis and social participation—was supported through the theoretical alignment of test items and subscales. Structural validity was further investigated using inter-scale correlations and CFA. Inter-scale correlations provided evidence of meaningful relationships among subscales, indicating coherence within the sensory processing domains. CFA results complemented these findings, confirming the intended underlying factor structure of the Arabic SPM-2 and highlighting areas needing further refinement, thereby collectively supporting the overall validity of the measure for the Arabic-speaking adult population. Each one of these is elaborated in detail below.

**Content validity** refers to the extent to which an assessment represents all facets of a particular construct it intends to measure. The content validity of the Arabic version of the SPM-2 was ensured through a rigorous process consistent with the original SPM-2 validation procedures. Specifically, the Arabic SPM-2 captures a comprehensive facet structure combining sensory domains (e.g., vision, hearing, and touch) and vulnerability dimensions (e.g., over-reactivity and under-reactivity), as originally conceptualized by Ayres Sensory Integration theory. Items included in the Arabic SPM-2 were first carefully translated and adapted to reflect cultural and linguistic contexts relevant to Arabic-speaking populations, particularly adults from Saudi Arabia. Subsequently, the translated items underwent expert scrutiny by a panel consisting of specialists in speech–language pathology, psychology, linguistics, and psychometrics. These experts evaluated the clarity, relevance, and representativeness of each item to ensure alignment with theoretical constructs and cultural appropriateness. Finally, statistical analyses, including reliability testing and confirmatory factor analysis, were conducted to confirm that retained items represent the full spectrum of sensory processing facets while maintaining desirable psychometric properties. Thus, similar to its original counterpart, the Arabic SPM-2 demonstrates robust content validity, accurately reflecting sensory integration constructs within an Arabic-speaking adult population.

**Construct validity** refers to how effectively an assessment measures the theoretical construct it aims to evaluate. For the Arabic version of the SPM-2, construct validity was rigorously established following procedures consistent with the original SPM-2 validation methodology. Specifically, the SPM-2, guided by Ayres Sensory Integration theory, assesses the primary theoretical constructs of interest including processing within sensory domains such as visual, auditory, tactile, olfactory/gustatory, proprioceptive, and vestibular systems, alongside praxis (motor planning and ideation) and social participation. Items from the original scale were translated and culturally adapted, and then carefully evaluated to ensure they reliably represent these underlying theoretical dimensions. The CFA and inter-scale correlations were conducted to statistically validate these constructs within the Arabic context, providing empirical support for structural relationships among sensory domains. Consequently, the construct validity of the Arabic SPM-2 builds upon its content validity and includes structural validity assessment, aligning closely with the original instrument’s theoretical foundation and validation process.

**Structural validity** refers to the extent to which the items of an assessment accurately reflect the intended underlying constructs or factors that the measure aims to evaluate. In this study, structural validity of the SPM-2 was assessed by examining both inter-scale correlations and CFA. First, inter-scale correlation analysis evaluated the strength and direction of relationships among the subscales, indicating the degree of coherence among sensory processing domains. Second, CFA was employed to test the factor structure proposed by the original measure, providing evidence of how well the observed data fit the hypothesized model. Inter-scale correlations revealed significant positive relationships among all sensory subscales and the Sensory Total scale (all correlations *p* < 0.001). Correlations among the six sensory subscales (Vision, Hearing, Touch, Taste and Smell, Body Awareness, and Balance and Motion) ranged from moderate (0.31 between Body Awareness and Hearing) to strong (0.69 between Balance and Motion and Body Awareness), supporting conceptual consistency across sensory dimensions. The sensory subscales strongly correlated with the Sensory Total score (range: from 0.59 for Hearing to 0.85 for Body Awareness), further confirming the internal coherence of the measure. The higher-order integrative scales, Planning and Ideas and Social Participation, exhibited moderate correlations with sensory subscales (ranging from 0.49 to 0.56 and 0.29 to 0.37, respectively), reinforcing their related but distinct roles within the sensory processing framework. The CFA results supported these findings, suggesting adequate fit indices and overall coherence among the factors. Together, these analyses affirm the structural validity of the Arabic SPM-2, establishing its effectiveness in measuring distinct yet interrelated sensory processing dimensions among Saudi adults.

### 3.3. Confirmatory Factor Analysis

The second step toward validating the Arabic version of the SPM-2 involved conducting a confirmatory factor analysis (CFA) to further explore its psychometric properties. The CFA assessed the factor structure initially suggested by the reliability analysis. The model yielded a significant chi-square value, χ^2^ (3052) = 4147.4 (*p* < 0.001), indicating some degree of model misfit, which is common with large sample sizes. However, additional fit indices suggested adequate model fit: RMSEA = 0.030 (90% CI [0.028, 0.032]); CFI = 0.74; and TLI = 0.73. Although the incremental indices (CFI = 0.74; TLI = 0.73) are below the 0.90 heuristic often cited for smaller models, simulation studies demonstrate that values in the 0.70s are acceptable in large-df models when accompanied by a small absolute-fit index (RMSEA ≤ 0.05) and theoretically coherent loadings [73]. Factor loadings were generally significant (*p* < 0.001), with standardized estimates predominantly ranging from 0.14 to 0.55, reflecting moderate-to-strong item–factor associations. Some items showed weaker or non-significant loadings, particularly within the Hearing and Social Participation subscales (e.g., Items 12, 14, 17, 71, 72, and 80), which aligns with their lower Cronbach’s α values identified earlier (0.43 and 0.18, respectively). Factor covariance analysis indicated substantial correlations between factors, ranging from 0.84 to 1.16 (*p* < 0.001), reinforcing the interconnected nature of sensory processing constructs measured by the SPM-2. Overall, the CFA results support the reliability findings, confirming the internal coherence and reinforcing the psychometric robustness of the Arabic SPM-2, while highlighting areas requiring further refinement for optimal diagnostic accuracy. The results reinforce the reliability analysis findings, supporting the internal coherence and psychometric robustness of the Arabic SPM-2, while identifying areas for further refinement.

An analysis of residual variances provided additional insights into the quality of the SPM-2 items. Generally, most residual variances remained below 0.6, indicating that the majority of items effectively captured the variance associated with their respective factors. However, some exceptions were noted, particularly within items from the Hearing and Social Participation subscales, which exhibited higher residual variances. These elevated residuals suggest that certain items within these subscales may not fully or accurately reflect their intended constructs and could benefit from revision or removal to enhance the scale’s overall measurement accuracy and reliability.

Table 3 summarizes the factor loadings from the CFA for each subscale of the Arabic SPM-2. Generally, most items across the subscales demonstrated statistically significant factor loadings (*p* < 0.05), suggesting that the items reliably measure the intended latent constructs and possess adequate convergent validity. The Vision, Touch, Taste and Smell, Body Awareness, Balance and Motion, and Planning and Ideas subscales had all items significantly contributing, indicating strong coherence within these subscales. However, the Hearing subscale had only 7 out of 10 items significantly loading, while the Social Participation subscale exhibited a notable issue, with only 5 out of 10 items showing significant loadings, including some negative loadings. This finding highlights substantial measurement challenges and potential content issues within the Social Participation subscale.

The covariances between factors were tested and showed substantial and statistically significant inter-factor relationships, with covariance estimates ranging from 0.84 to 1.16 (all *p*-values < 0.001). This finding indicates strong correlations among sensory processing domains measured by the SPM-2, supporting the theoretical assumption that sensory processing functions are closely interconnected. These high covariances suggest robust internal consistency across factors and affirm the SPM-2’s capability to evaluate a broad spectrum of sensory processing characteristics simultaneously.

The path diagram in Figure 1 illustrates the CFA model for the Arabic SPM-2, visually depicting the relationships among the eight subscale factors (Vision, Hearing, Touch, Taste and Smell, Body Awareness, Balance and Motion, Planning and Ideas, and Social Participation) and their corresponding items. Arrows from factors to items represent factor loadings, indicating the degree to which each item is influenced by its associated latent construct. Curved lines among factors reflect covariance relationships, indicating that sensory processing constructs are strongly interrelated. The diagram reinforces CFA findings by visually emphasizing the coherence and interconnectedness of the measured sensory processing domains.

### 3.4. Sensory Processing Differences by Sex and Age Category

A two-way ANOVA (Sex × Age) tested group differences on the eight domain scores and the Sensory Total. Age emerged as the only consistent determinant of performance. Vision, Touch, Taste–Smell, Social Participation, and the Sensory Total all varied significantly across the three age groups (Vision F(2, 393) = 4.48; *p* = 0.012; Touch F = 4.51; *p* = 0.012; Taste–Smell F = 3.74; *p* = 0.025; Social Participation F = 4.95; *p* = 0.008; Sensory Total F = 4.72; *p* = 0.009), with small but meaningful effects (partial η^2^ = 0.019–0.024).

By contrast, sex contributed no unique variance to any subscale (largest F = 1.15; *p* = 0.284, partial η^2^ ≤ 0.003), and no Sex × Age interactions reached significance (largest F = 2.45; *p* = 0.088). Thus, the age-related patterns were parallel for men and women. Four domains—Hearing, Body Awareness, Balance and Motion, and Planning and Ideas—showed neither age nor sex effects (all ps > 0.05).

In summary, age modulated five of nine sensory measures, whereas sex did not influence any. The data suggest that visual, tactile, chemosensory, and social-participation capacities—as well as the global Sensory Total—decline modestly across adulthood, while auditory, proprioceptive, vestibular, and praxis-related functions remain comparatively stable. Because age effects were small but pervasive and consistent across sexes, future screening or intervention efforts in Arabic-speaking adults may benefit from age-stratified norms and from tailoring programmes to the sensory domains most susceptible to age-related change. At the same time, the absence of sex differences supports the use of a single, unified scoring rubric for men and women in clinical and research applications.

### 3.5. Diagnostic Feature of the Arabic SPM-2

Because we suspected that responses of 2 (Frequently) or 3 (Always) indicate participants at risk of SPDs, we examined the ability of the Arabic SPM-2 to detect at-risk populations in our sample (see Figure 2).

Before conducting inferential tests, we explored the distribution of responses to understand patterns in participant ratings. Our primary goal was to identify the proportion of participants selecting 2 (Frequently) or 3 (Always), check for skewness (a clustering of responses toward higher scores), and detect outliers (participants who consistently rated items as high). The findings revealed that certain items, such as 31, 71, 23, and 24, had the highest percentage of participants selecting 2 or 3, suggesting frequent sensory-related behaviors. Skewness analysis indicated a tendency for responses to cluster toward the higher end of the scale, implying non-random response patterns. Additionally, multiple participants displayed consistently high ratings across items, highlighting the need for further investigation into potential sensory processing challenges among these individuals.

To better understand response patterns, we conducted a cluster analysis to determine whether specific subgroups consistently reported high scores. The primary objective of conducting this cluster analysis was to identify distinct groupings of individuals based on their sensory processing responses and, thereby, uncover key patterns and variability within the dataset. The results revealed four clearly differentiated clusters, with a relatively balanced distribution—Cluster 1 (135 participants), Cluster 2 (36 participants), Cluster 3 (86 participants), and Cluster 4 (142 participants). The analysis of the sum of squares indicates that Clusters 1 (9183) and 4 (10169) demonstrate higher within-cluster variability, while Clusters 2 (3109) and 3 (4974) show comparatively lower variability. K-means grouped participants into four clusters: moderate responders (n = 135), low responders (n = 36), mildly hypersensitive (n = 86), and highly responsive (n = 142).

Cluster 1 reflects individuals reporting moderate sensory responsiveness, exhibiting stable yet slightly elevated scores across most items, suggestive of typical processing with minor fluctuations. By contrast, Cluster 2 is characterized by consistently negative centroid values, indicative of low responsiveness or under-reactivity. Cluster 3 captures mild hypersensitivity, with negative item deviations that are not as pronounced as those observed in Cluster 2; it may thus reflect participants experiencing some sensory difficulties but not severe impairments. In turn, Cluster 4 exhibits overall positive centroid values, suggesting heightened responsiveness or hyper-reactivity that may place these individuals at greater risk of sensory processing challenges.

The inspection of mean trends across the four groups supports these interpretations, showing comparatively stable responses for Clusters 1 and 3, marked negative deviations for Cluster 2, and pronounced positive deviations for Cluster 4. This heterogeneity underscores the multi-dimensional nature of sensory processing, whereby certain individuals experience typical responses, whereas others exhibit either heightened or attenuated sensitivity. The cluster plot corroborates these findings, revealing a clear separation that points to structured variability rather than random fluctuations among participants (See Figure 3). Notably, Cluster 2 appears uniquely distant, hinting at more pronounced atypicalities in sensory functioning. Future analyses, such as ANOVA or logistic regression, may elucidate potential associations with demographic variables (e.g., age or sex), while further investigation (e.g., PCA or factor analysis) could identify the specific items that drive these cluster distinctions, offering deeper insights into the underlying mechanisms of sensory processing.

## 4. Discussion

The results of this study align well with prior research conducted on the original SPM and SPM-2 instruments across diverse cultural contexts. Lai et al. (2011) validated the Chinese version of the SPM and highlighted similar psychometric strengths, underscoring its cross-cultural adaptability [38]. Moreover, the validation of the Arabic Preschool and Home versions of the SPM by Alkhalifah et al. [57] yielded comparable reliability and validity results, supporting the broader applicability of the measure within Arabic-speaking populations. Additionally, a study conducted in Italy by Narzisi et al. [74] used the SPM-2 to describe sensory profiles in school-aged children with autism spectrum disorder. The findings confirmed the tool’s sensitivity in identifying distinct sensory processing patterns, supporting its use in clinical assessment and intervention planning across cultural settings. This further demonstrates the SPM-2’s international applicability and relevance for both research and practice in diverse populations.

Importantly, this study also incorporated specific analyses aimed at the early detection of sensory processing risks. Through descriptive and frequency analyses, it was observed that a notable portion of the sample selected higher frequency responses (Frequently or Always) across several sensory domains. This finding suggests the presence of significant sensory processing challenges even within a general adult population. Cluster analysis further reinforced this, identifying distinct subgroups exhibiting consistent and elevated sensory processing difficulties. Such analytical approaches facilitate the identification of individuals potentially at risk, aligning with similar methodological approaches advocated in the previous literature [75,76].

The findings on structural validity, confirmed by CFA, are consistent with previous work by Miller-Kuhaneck et al. [22], who established the theoretical consistency of sensory processing constructs assessed by the SPM. Specifically, the moderate-to-strong correlations among sensory subscales and their association with the overall Sensory Total support Ayres’ Sensory Integration theory, which emphasizes the interconnectedness of sensory systems. However, certain subscales such as Social Participation and Hearing demonstrated lower internal consistency and weaker factor loadings, a challenge also noted by recent international adaptations [77]. These outcomes suggest that specific items within these subscales may not fully resonate with the cultural interpretations of sensory experiences in the Saudi context and may require further refinement.

Convergent validity, an essential component of construct validation, has previously been established between the SPM-2 and other sensory assessments, such as the Sensory Profile-2 (SP2) and the Adolescent/Adult Sensory Profile (A/ASP). Skocic and Reed demonstrated significant correlations between the SPM-2 Adult form and the A/ASP, reinforcing the scale’s construct validity [77]. Similarly, Jones et al. found significant correlations between school-age versions of the SPM-2 and the SP2, emphasizing the SPM-2’s consistent psychometric performance across age groups and contexts [76].

A noteworthy finding of this research is the significant influence of age on sensory processing scores. Older participants reported increased sensory processing difficulties across multiple domains compared to younger adults, a finding echoed by Brown et al. [58], who indicated age-related variability in sensory responsiveness using the SPM-2. Conversely, the lack of significant sex differences in sensory processing observed in this study aligns with findings reported by Lai et al. [38] and Hansen and Jirikowic [78], suggesting that age may be a more critical factor than sex in determining sensory processing variability in adults.

In summary, this research contributes significantly to the validation of the Arabic SPM-2 Adult form, providing robust evidence of its reliability, validity, and clinical utility in identifying sensory processing difficulties among Saudi adults. These findings not only strengthen the existing psychometric literature but also highlight the importance of culturally adapted assessments in accurately capturing sensory processing experiences across diverse populations.

## 5. Limitations

Despite the robust psychometric findings, certain limitations warrant consideration. One primary limitation is the relatively low reliability of specific subscales, particularly Social Participation and Hearing, which is consistent with findings from Bäckström et al. [79]. These lower internal consistency scores might reflect cultural or linguistic differences affecting the interpretation of social interaction or auditory processing items. Additionally, while descriptive and cluster analyses provided valuable insights into potential at-risk populations, further clinical validation would be beneficial to confirm the diagnostic and predictive utility of these subgroups. The present study did not include an external gold-standard measure; consequently, the instrument’s diagnostic accuracy and prognostic value remain to be tested in clinical samples. The volunteer (non-probability) sample limits external generalizability and precludes prevalence estimates; therefore, results should be interpreted strictly as evidence of internal psychometric performance. We did not stratify results by other sociodemographic characteristics (e.g., education or residence) because several categories had small cell counts; future studies with larger, probabilistic samples should examine these factors. Accordingly, we refrain from making clinical recommendations until such criterion-based evidence is available.

## 6. Future Directions

Future research directions include examining test–retest reliability, responsiveness to sensory-based interventions, and further cross-cultural comparative studies involving other Arabic-speaking countries. Given the identified limitations in specific subscales, future studies might also explore item-level revisions or adaptations to enhance cultural and psychometric relevance.

## 7. Theoretical and Practical Implications

Nonetheless, this study has several practical and theoretical implications. From a screening perspective, the instrument’s strong internal consistency (α = 0.89) and replicable eight-factor structure indicate that it can generate reliable sensory profiles for Arabic-speaking adults. However, definitive diagnostic or prognostic use will require future studies that examine criterion validity—e.g., correlations with functional outcomes or receiver-operating-characteristic (ROC) analyses in clinical samples—before full clinical utility can be claimed [80]. Theoretically, these findings support the continued relevance of sensory integration theories and highlight the necessity for culturally sensitive adaptations of widely used sensory assessment tools.

## 8. Conclusions

This study provides strong evidence supporting the reliability, validity, and clinical utility of the Arabic version of the SPM-2 Adult Self-Report form for use in Saudi Arabia. Overall, the Arabic SPM-2 demonstrated robust internal consistency, structural and construct validity, and sensitivity to age-related differences, underscoring its appropriateness for capturing sensory processing patterns among Arabic-speaking adults. Notably, the inclusion of frequency-based descriptive analysis and cluster analysis in this study facilitated the early identification of adults potentially at risk of SPDs. Such analytic strategies proved valuable for screening and could guide clinical interventions and preventive measures in occupational therapy, psychological services, and educational contexts.

Although certain subscales—especially Social Participation and Hearing—require further cultural and psychometric refinement, the Arabic SPM-2 effectively addresses the critical need for culturally adapted sensory assessment tools within Arabic-speaking populations. Continued research exploring test–retest reliability, responsiveness to sensory interventions, and further cross-cultural comparisons is recommended. Overall, this study significantly contributes to the field by extending the applicability of Ayres Sensory Integration assessments into diverse linguistic and cultural contexts, thereby enhancing early diagnostic capabilities and therapeutic outcomes for SPDs in adults.

## Figures and Tables

**Figure 1 jcm-14-03283-f001:**
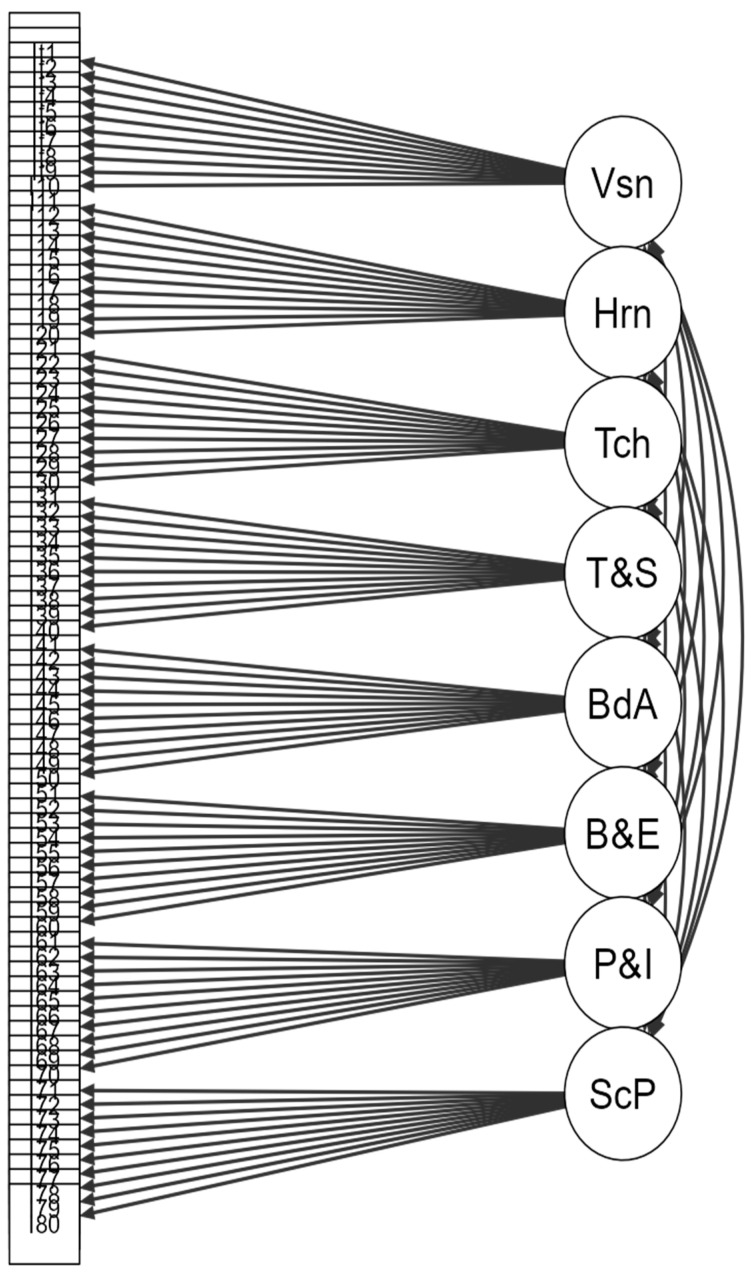
Path diagram for factor analysis of Arabic SPM-2.

**Figure 2 jcm-14-03283-f002:**
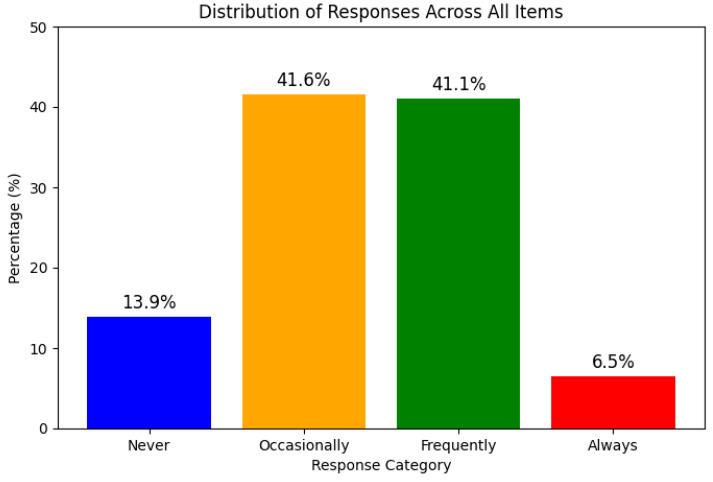
Distribution of responses across all items by frequency.

**Figure 3 jcm-14-03283-f003:**
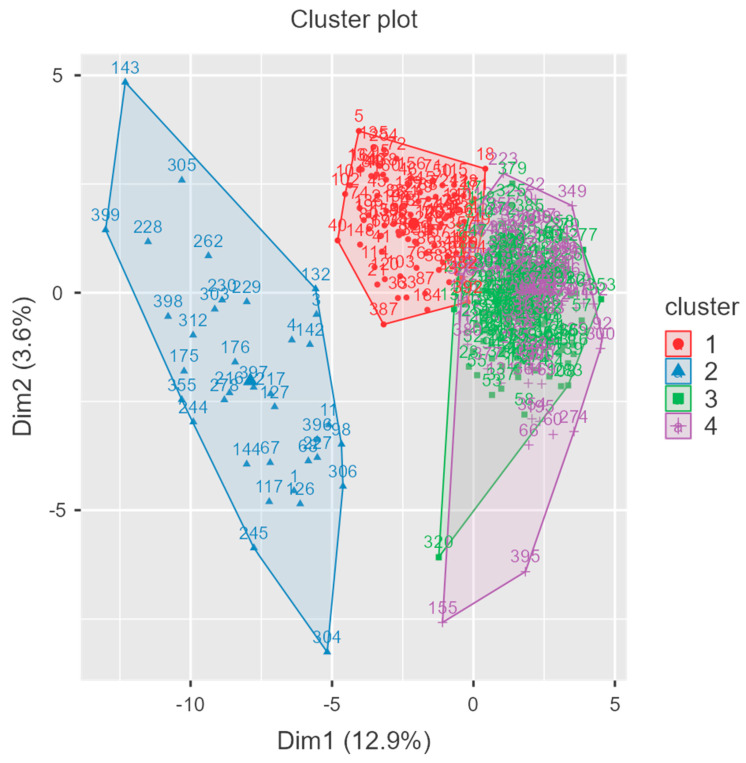
Cluster plot for at-risk diagnostic feature of Arabic SPM-2.

**Table 1 jcm-14-03283-t001:** Demographic characteristics of participants (N = 399).

Sex	Age Category	n	M (SD) for Items (Range: 0–3)
Female	Young Adult	50	1.29 (0.70)
	Middle-Aged	109	1.40 (0.75)
	Old	46	1.48 (0.74)
Male	Young Adult	50	1.40 (0.73)
	Middle-Aged	106	1.42 (0.74)
	Old	38	1.51 (0.74)

Note: Values represent overall means and standard deviations across the 80 items for each subgroup. Item responses range from 0 (no difficulty) to 3 (extreme difficulty). All participants were from Saudi Arabia with no reported history of psychiatric conditions.

**Table 2 jcm-14-03283-t002:** Reliability analysis of SPM-2 subscales and overall scale.

Subscale	Mean	SD	Cronbach’s α
Vision	1.3542	0.33265	0.55
Hearing	1.4347	0.31355	0.43
Touch	1.4411	0.36432	0.58
Taste and Smell	1.4223	0.31998	0.52
Body Awareness	1.3547	0.39442	0.70
Balance and Motion	1.3355	0.35855	0.58
Planning and Ideas	1.3438	0.35634	0.61
Social Participation	1.3670	0.25598	0.18
Overall SPM-2 Scale	1.3783	0.24758	0.89

**Table 3 jcm-14-03283-t003:** Factor loadings for confirmatory factor analysis of SPM-2.

Factor	Range of Factor Loadings	Significant Indicators
Vision	0.16–0.36	10/10
Hearing	0.00–0.36	7/10
Touch	0.13–0.39	10/10
Taste and Smell	0.10–0.40	10/10
Body Awareness	0.24–0.40	10/10
Balance and Motion	0.15–0.39	10/10
Planning and Ideas	0.11–0.38	10/10
Social Participation	−0.03–0.37	5/10

## Data Availability

The data presented in this study are available on request from the first author.

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
