# Peer review of "Diagnostic and Psychometric Properties of the Arabic Sensory Processing Measure—Second Edition, Adult Version"

_jcm, 2025, doi:10.3390/jcm14103283_

Round 1

Reviewer 1 Report

Comments and Suggestions for Authors

This is an interesting article. But I see two issues:

1) Article is really too long. I feel that introduction and resultsm are double as long as they should be. Readers become tired when read so long textes. please shorten yourt article.

2) Authors should check which statistical values are important for interpretation and which aren't .Table 2 can be replaced with text as nothing but p value is really important here. The same with table 4. Authors present several tablöes after Table 4 calling them Table 1.  For example, Table 1. Two-Way ANOVA Results for Sensory Processing Subscales by Sex and Age 597
Category (N = 399).    This is probably Table 5, right? What is reklevant here? P value is relevant; isd DF relevant? F value? I think no. Please provide besser less than more, and only values which can be interpreted.

Reviewer 2 Report

Comments and Suggestions for Authors

I have reviewed the topic, namely, Diagnostic and Psychometric Properties of the Arabic Sensory Processing Measure-Second Edition, Adult Version.

My observations are as follows:

Major observations

Introduction

  1. I would suggest to provide a more detailed interpretation of the Sensory processing disorders (SPD).
  2. I would recommend that the Authors could improve the purification of the topic and explain why is it so important to manage the symptoms of SPD.
  3. It seems necessary to reveal the novelty and relevance of this study.
  4. Why is this study particularly relevant from a clinical medical point of view?
  5. At the end of the Introduction, I suggest that the Authors must write down the clear aim of this study.
  6. Overall, the Introduction is too long and confusing. It seems necessary to make major revisions.

Methods

  1. Information on lines 248-253 and on lines 258-266 overlaps completely.
  2. Line 211: The „Sample“ section must contain, as follows:

8.1. How was a representative sample size calculated?

8.2. What selection techniques have been applied to the participants in the study?

8.3. What were the criteria for inclusion and exclusion of study participants?

8.4. The results of the study should not be included in the methods section, too.

  1. Lines 298-314: the provided information should be shortened and structured.

Results

  1. The same inconsistencies were observed. I propose that the Authors should provide only the results of this study and do not write unnecessary information (for example, on lines 347-355).
  2. Whilst factorial analysis is performed, there is no need to provide results obtained from correlation analysis.
  3. Generally, the Results section must be overwritten. The point is that the Authors have provided all possible calculations. It seems appropriate to structure the substantive results and describe them more briefly.
  4. Furthermore, Figure, namely, Path Diagram for Factor Analysis of Arabic SPM-2 should be revised by explicitly specifying the relationships between variables were analysed. The current image format renders nothing.

Discussion

  1. Lines 660-671: This information is not a discussion. I recommend that the Authors do not duplicate the same information in the manuscript.
  2. Although the Author's aimed to validate the questionnaire, it is necessary to submit a translated questionnaire as a supplemental material at the end of the manuscript.

Minor observations

  1. Keywords should be arranged in alphabetical order.
  2. APA quotation style should be changed to the MDPI or ACS style.
  3. Line 80: “This Framework conceptualizes Sensory processing Along two Continua: …”. I suggest that the Authors could clarify the sentence.

Reviewer 3 Report

Comments and Suggestions for Authors

Major Issues:

  • Line 30: Are the reported values of CFI = 0.74 and TLI = 0.73 acceptable? These fall below commonly accepted thresholds and should be further justified or addressed.

  • Line 35: The claim that the Arabic SPM-2 is “clinically useful” lacks empirical support. Since criterion validity was not assessed, no conclusions can be drawn regarding the tool's prognostic or diagnostic utility.

  • Lines 181–184: The limitations of previously adapted tools for the Arabic-speaking population were not adequately discussed in the Introduction. This omission weakens the narrative used to justify the need for the present study.

  • The Methods section needs to be reorganised for improved clarity and readability. Additional subheadings are recommended (e.g., Sampling strategy and recruitment, Inclusion/exclusion criteria, etc.).

  • If the authors classify this study as “cross-sectional” (Line 248), then the STROBE guidelines for cross-sectional studies must be strictly followed, particularly when reporting methods. However, I question whether this work fits the cross-sectional classification, as it does not measure prevalence and uses a non-probabilistic sampling strategy.

  • Lines 248–266: This section contains redundant text and should be revised for conciseness.

  • There is no information provided regarding the sample size calculation, which is essential for assessing the study’s statistical validity.

  • Tables 2 and 4 appear redundant and do not add meaningful value; they can be omitted.

  • Figure 2 is not legible and should be reformatted or removed.

  • The Results section is overly verbose. Since the numerical data are already presented in the tables, the textual description should be streamlined to highlight only key findings and interpretations.

  • The ordering of tables and figures is inconsistent and should be corrected to follow the sequence in the text.

  • The analyses were not stratified by key sociodemographic variables beyond age and sex. Stratified analyses could have provided more nuanced insights.

  • The Discussion section does not sufficiently support the claim of clinical utility.

  • The Limitations subsection fails to discuss the implications of the sampling strategy, particularly its impact on generalisability.

Minor Issues:

  • Line 27: The abbreviation “ST” appears without prior definition.

  • Line 44: The term “SPD” is introduced here but is not used consistently throughout the text.

  • Line 143: The term “bibliometric analyses” would be more appropriate than “scientometric analyses”. Scientometrics typically concern evaluative indicators (e.g., h-index, impact factor), while bibliometrics deals more with publication patterns, collaboration networks, and citation analysis.

  • Lines 325–344: The names of R packages should be italicised, in accordance with academic style conventions.

  • The reference style does not adhere to the journal’s author guidelines and must be corrected accordingly.

  • The Interscale Correlation Matrix table should not include asterisks (***) as all p-values are < 0.001. Instead, this information can be stated once in the table note.

Round 2

Reviewer 1 Report

Comments and Suggestions for Authors

-

Reviewer 3 Report

Comments and Suggestions for Authors

The authors did a great job responding to all my prior comments and adjusted the manuscript accordingly. I have no more comments to make. The manuscript is in good shape now.